# Improved Survival after Transarterial Radioembolisation for Hepatocellular Carcinoma Gives the Procedure Added Value

**DOI:** 10.3390/jcm11247469

**Published:** 2022-12-16

**Authors:** Cristina Mosconi, Alberta Cappelli, Cinzia Pettinato, Maria Adriana Cocozza, Giulio Vara, Eleonora Terzi, Maria Cristina Morelli, Elisa Lodi Rizzini, Matteo Renzulli, Francesco Modestino, Matteo Serenari, Rachele Bonfiglioli, Letizia Calderoni, Elena Tabacchi, Matteo Cescon, Alessio Giuseppe Morganti, Franco Trevisani, Fabio Piscaglia, Stefano Fanti, Lidia Strigari, Alessandro Cucchetti, Rita Golfieri

**Affiliations:** 1Department of Radiology, IRCCS Azienda Ospedaliero-Universitaria di Bologna, 40138 Bologna, Italy; 2Fondazione IRCCS Ca’ Granda Ospedale Maggiore Policlinico, 20122 Milan, Italy; 3Division of Internal Medicine, IRCCS Azienda Ospedaliero-Universitaria di Bologna, 40138 Bologna, Italy; 4Division of Internal Medicine for the Treatment of Severe Organ Failure, IRCCS Azienda Ospedaliero-Universitaria di Bologna, 40138 Bologna, Italy; 5Radiation Oncology, IRCCS Azienda Ospedaliero—Universitaria di Bologna, 40138 Bologna, Italy; 6General Surgery and Transplantation Unit, IRCCS Azienda Ospedaliero—Universitaria di Bologna, 40138 Bologna, Italy; 7Nuclear Medicine Unit, IRCCS, Azienda Ospedaliero—Universitaria di Bologna, 40138 Bologna, Italy; 8Department of Medical and Surgical Sciences, Semeiotica Medica, IRCCS Azienda Ospedaliero—Universitaria di Bologna, 40138 Bologna, Italy; 9Department of Medical Physics, IRCCS Azienda Ospedaliero-Universitaria di Bologna, 40138 Bologna, Italy; 10Department of Medical and Surgical Sciences—DIMEC, Alma Mater Studiorum—University of Bologna, 40138 Bologna, Italy; 11Department of General Surgery, Morgagni—Pierantoni Hospital, 47121 Forlì, Italy; 12Alma Mater Studiorum, Università Degli Studi Di Bologna, 40138 Bologna, Italy

**Keywords:** radioembolization, hepatocellular carcinoma, HCC

## Abstract

Background: Transarterial Radioembolisation (TARE) requires multidisciplinary experience and skill to be effective. The aim of this study was to identify determinants of survival in patients with hepatocellular carcinoma (HCC), focusing on learning curves, technical advancements, patient selection and subsequent therapies. Methods: From 2005 to 2020, 253 patients were treated. TARE results achieved in an initial period (2005–2011) were compared to those obtained in a more recent period (2012–2020). To isolate the effect of the treatment period, differences between the two periods were balanced using “entropy balance”. Results: Of the 253 patients, 68 were treated before 2012 and 185 after 2012. In the second period, patients had an Eastern Cooperative Oncology Group (ECOG) Performance Status (PS) score of 1 (*p* = 0.025) less frequently, less liver involvement (*p* = 0.006) and a lesser degree of vascular invasion (*p* = 0.019). The median overall survival (OS) of patients treated before 2012 was 11.2 months and that of patients treated beginning in 2012 was 25.7 months. After reweighting to isolate the effect of the treatment period, the median OS of patients before 2012 increased to 16 months. Conclusions: Better patient selection, refinement of technique and adoption of personalised dosimetry improved survival after TARE. Conversely, sorafenib after TARE did not impact life expectancy.

## 1. Introduction

Hepatocellular carcinoma (HCC) is the sixth most common cancer and the fourth leading cause of cancer-related death worldwide [1]. Only approximately 30–40% of patients can be treated with any hope of recovery [2,3]. The remaining patients undergo locoregional or systemic therapies, of which transarterial chemoembolisation (TACE) is the most frequent approach [4,5]. Unfortunately, the efficacy of TACE is poor in the treatment of large (>5 cm) and multinodular tumours [6,7,8,9]. Furthermore, the presence of macroscopic neoplastic vascular invasion (MaVI) is generally considered a contraindication to this intra-arterial treatment [10].

In the presence of large HCCs with or without MaVI, transarterial radioembolisation (TARE) represents an intra-arterial therapy alternative to TACE or to systemic therapies and has been shown to be effective in inducing tumour necrosis and delaying its progression, ultimately leading to a survival benefit [11,12,13]. Unfortunately, to date, the lack of elevated levels of clinical evidence has not allowed TARE to be definitively placed in the international guidelines [4,14,15]. This is especially true when TARE is compared to systemic therapies [14,15]. Two recent randomised controlled trials (RCTs), the SARAH [14] and the SIRveNIb studies [15], failed to show a survival benefit when compared to sorafenib in patients with advanced HCC. However, this does not mean that TARE should not have a place within the guidelines, especially considering the difficulties and the potential distortions capable of undermining those studies which compare an interventional procedure (TARE) with a drug (sorafenib) [16]. Moreover, in the latest update of the Barcelona Clinic Liver Cancer (BCLC) staging system in 2022, TARE was included as a therapy in BCLC 0 and A when other treatments, such as ablation, resection, and TACE, were not feasible [17,18].

Transarterial radioembolisation is a sophisticated technical procedure requiring great skill and expertise, together with multidisciplinary management, involving various medical and non-medical personnel, involving a significant learning curve. In both the above-mentioned RCTs, the experience of the participating centre was highly heterogeneous, highlighting several limits to TARE application, thus, artificially reducing its effectiveness [19]. Lack of experience may have influenced the selection of patients, which was not entirely correct, as well as having negatively influenced the significant impact of dosimetry. These aspects led to median survival rates lower (8.0 months in SARAH and 8.8 months in SIRveNIb) than those reported in several retrospective large cohort studies involving highly experienced centres [11,12,13].

The aim of the present paper was to evaluate the evolution of TARE, the impact of the learning curve in identifying the best clinical conditions determining survival (i.e., improvement in patient selection), technical progress including dosimetry, and the effect of combining systemic therapies.

## 2. Materials and Methods

The present study was a single-centre study, retrospectively analysing a prospectively collected database beginning in 2005. From 2005 until December 2020, a total of 373 patients were treated with TARE at the Authors’ institution. The present analysis focused on the 253 patients treated for HCC. The study conformed to the ethical guidelines of the Declaration of Helsinki, and the data collection and analyses were approved by the Institutional Review Board of the centre. All patients provided informed written consent for the processing of personal data according to Italian Data Protection Authority laws (legislative decrees n. 196 of 2003 and n. 101 of 2018).

### 2.1. Patient Selection

The patients treated with TARE were those deemed not amenable to curative or TACE treatments by the local multidisciplinary team. The decision was commonly dictated by the presence of locally advanced HCC with MaVI, hypovascular-infiltrating HCC, very large tumours, and/or whenever previous therapies failed to control the disease. The diagnoses of HCC and cirrhosis were based on histology or the non-invasive criteria obtained during the study period [4,5]. The patients were enrolled from September 2005 (when this treatment became available in the Authors’ department) until December 2020. Detailed selection criteria are provided in the Appendix A. Briefly, the main selection criteria were an Eastern Cooperative Oncology Group (ECOG) Performance Status (PS) ≤ 1, a Child—Pugh score ≤ 8, and serum bilirubin ≤ 2 mg/dL.

### 2.2. Evolution of the TARE Procedure

Since 2005, TARE has been performed using 90Y resin microspheres (SIR-Spheres; Sirtex Medical Limited, Sydney, Australia) [20]. The procedure always started with a hepatic angiography to identify the arteries feeding the tumour. The lobar approach carried out at the beginning of the study was progressively substituted by segmental or bisegmental catheterisation. The patients were then injected with 150 MBq of 99mTc-macroaggregated albumin (MAA) through an angiographic catheter.

Since 2005, MAA has been used to calculate the hepatopulmonary shunt fraction and to evaluate other eventual extra-hepatic uptakes, using planar scintigraphy imaging. The activity delivered was, however, calculated using the body surface area (BSA) formula provided by the manufacturer which did not consider the doses absorbed by the tumour, liver, and lung.

Starting in January 2012, immediately after undergoing planar scintigraphy, each patient underwent a single photon emission computed tomography (SPECT) scan to additionally confirm the correct MAA distribution in the lesion. This was done to visualise the distribution of the particles in the liver, as a pretreatment simulation study, and to allow for calculating the tumour-to-liver ratio (TLR) necessary for a subsequent partition model. Provisional dosimetry was then carried out following Medical Internal Radiation Dose (MIRD) formalism in order to verify that the activity calculated using the BSA formula provided an absorbed dose to the tumour of at least 100–120 Gy, together with cautiously absorbed doses to the entire normal liver and to the lungs of below 30–40 Gy and 10–15 Gy, respectively.

After 1 to 2 weeks, the patients were implanted with 90Y resin microspheres. Until December 2011, their final distribution was verified using a post-treatment 90Y Bremsstrahlung SPECT scan; subsequently, it was reached using a 90Y positron emission tomography/computed tomography (PET/CT) scan.

### 2.3. Adverse Events, Follow-Up, and Subsequent Therapies

Imaging and laboratory tests were repeated at 1, 3, and 6 months, and were subsequently scheduled by the referring physician. Adverse events (AEs) were prospectively recorded and herein redefined according to the National Cancer Institute’s Common Terminology Criteria for Adverse Events (NCI-CTCAE) version 3.0 [21]. Detailed definitions are reported in the Appendix A.

On the basis of the clinical and radiological follow-ups, eventual subsequent therapies were decided upon by the local multidisciplinary team, including resection and transplantation. When appropriate downstaging was obtained, additional locoregional therapies, as well as the adoption of systemic therapies, were carried out, mainly on the basis of the perceived probability of clinical success and available alternatives.

### 2.4. Statistical Analysis

Common descriptive analyses were carried out using the chi-square, Fisher’s exact, and Kruskal–Wallis tests. To isolate the effect of the period of treatment, the differences observed in the clinical and radiological features between the two time periods were balanced using “entropy balance”. The latter is a data preprocessing procedure which allows reweighting of a dataset so that the covariate distributions in the reweighted data satisfy specified conditions. In other words, reweighting allowed estimating what would have happened if the patients treated at the beginning of the TARE study had been treated in the more recent era. The correct balancing was verified by means of standardised differences (d-values).

Overall survival (OS) was computed from the day of TARE to the day of the last follow-up visit or death. As previously stated, during this time interval, the patients eventually underwent additional therapies capable of modifying overall survival. Consequently, second therapies were recorded, and survival analyses were carried out using a time-dependent approach. Since different therapies, with different expected efficacies, were used, they were graded hierarchically from those considered potentially curative, namely, transplantation and resection, to TACE and, finally, to sorafenib [22]. The most curative therapy adopted was then considered. Survival was estimated using the Kaplan–Meier test and Cox regression with appropriate time-dependent approaches. All the statistical analyses were carried out using Stata 15.0 (StataCorp. 2017. Stata Statistical Software. StataCorp LLC, College Station, TX, USA). 

## 3. Results

A total of 253 patients underwent TARE during the study period. Of these, 68 (26.9%) were treated before January 2012 and 185 (73.1%) after this time point. The clinical features stratified by this temporal threshold are reported in Table 1.

### 3.1. Patient Selection

In the most recent period, the patients had an ECOG–PS score of 1 (*p* = 0.025) less frequently and were less frequently treated after TACE failure (*p* = 0.039) as compared to the first period. The percentage of livers involved by tumours was lower in the most recent period (*p* = 0.006) as was the quantity of activity delivered (*p* = 0.001). Finally, more recently patients were treated with a less advanced degree of MaVI, when present (*p* = 0.019), and more frequently, treated with sorafenib after TARE (*p* = 0.008).

### 3.2. Unweighted Efficacy

The median follow-up of patients treated before 2012 was 11.4 months (IQR: 5.7–23.9), counting 67 deaths (98.5%); the median follow-up of those treated from 2012 onward was 14.3 months (IQR: 6.5–26.3), counting 96 deaths (51.9%). The median OS of patients treated before 2012 was 11.2 months (95% CI: 5.6–23.8) and that of patients treated from 2012 onward was 25.7 months (95% CI: 10.3–50.7), providing an unadjusted hazard ratio (HR) of 0.48 (95% CI: 0.35–0.66) for the most recent period (*p* = 0.001).

### 3.3. Weighted Efficacy

Reweighting using entropy balance allowed isolating the effect of the period of treatment as the main clinical and radiological feature of the patients treated before 2012; the results were now almost identical to those of the more recent period (Table 2).

With balanced covariates, the median OS of the patients treated before 2012 increased to 16 months (IQR:.6.6–21.9). The median OS of the patients treated from 2012 onward remained 25.7 months, providing an adjusted HR of 0.53 (95% CI: 0.41–0.67) for the most recent period (Figure 1; *p* = 0.001).

In determining OS, the interaction effect between the reweighted clinical variables and the treatment period (Figure 2) suggested the improved efficacy of TARE for the majority of clinical situations, with the exception of patients belonging to ALBI class B/C (*p* = 0.194), those with an ECOG–PS score of 1 (*p* = 0.750), those with larger (*p* = 0.347) and bilobar tumours (*p* = 0.308) having a tumour burden of >13% of the total liver volume (*p* = 0.229). No improvement was observed in patients without MaVI (*p* = 0.153) or in those with portal vein 1 (Vp1) neoplastic invasion (*p* = 0.325).

### 3.4. The Effect of the Tumour-Adsorbed Dose

After reweighting, the median OS of patients without a tumour-adsorbed dose was the same as patients treated before 2012, thus, 16 months as previously reported. The median OS of patients with an adsorbed dose <120 Gy was 15.7 months (IQR: 5.6–34.1) and that of patients with an adsorbed dose ≥120 Gy was 26.0 months (IQR: 10.5–53.3). The HR of patients treated with <120 Gy as compared with those treated with an unmeasured dosage was 1.07 (95% CI: 0.54–2.11; *p* = 0.839). The HR of patients treated with ≥120 Gy as compared with those treated with an unmeasured dosage was 0.49 (95% CI: 0.31–0.77; *p* = 0.002); the HR of patients treated with ≥120 Gy as compared with those treated with <120 Gy was 0.53 (95% CI: 0.29–0.94; *p* = 0.033).

### 3.5. The Effect of Subsequent Therapies

In the reweighted population and with a time-dependent adjustment, only surgery provided a survival advantage as compared to the absence of additional treatment, with an +HR of 0.39 (95% CI: 0.20–0.74; *p* = 0.004). Neither TACE (HR: 0.81; 95% CI: 0.48–1.36; *p* = 0.436) nor sorafenib (HR: 0.85; 95% CI: 0.33–2.18; *p* = 0.741) added any benefit to that of TARE.

### 3.6. Safety

The safety profile before and after reweighting is reported in Table 3. Patients treated before 2012 had a higher occurrence of grade III-IV pulmonary events (*p* = 0.019), radiation-induced liver disease (RILD) (*p* = 0.016) and grade III-IV acute cholecystitis (*p* = 0.019), but less subjective grade I-II fatigue (*p* = 0.039) as compared to the patients treated in the second period. The estimation of how these morbidities would have changed with the modern approach, using reweighting, showed an absolute risk reduction (ARR) in grade III-IV pulmonary events of 4.4% (from 5.9 to 1.5%), an ARR for RILD of 4.8% (from 7.4 to 2.6%) and a zero score for grade III-IV acute cholecystitis (from 4.4 to 0%). The ARR for overall morbidity was 11.7% (from 33.8 to 22.1%).

## 4. Discussion

Conflicting results have been reported in the literature available regarding the use of TARE for HCC [11,12,13,14,15]; to date, more than one doubt has been raised regarding the design of the main RCTs, the poor results of which have closed the door to TARE in the treatment of intermediate/advanced HCC otherwise amenable to sorafenib [19,22,23,24]. The main criticisms regarded patient population selection which was not completely correct. Transarterial radioembolisation was particularly efficacious in patients with Vp1-Vp3 MaVI [20,25]; conversely, a consistent percentage of the RCT patients had Vp4, and subgroup analyses for this MaVI extension were not provided. The expertise in managing TARE was also highly heterogeneous in both the SIRveNIB trial (carried out at 11 centres of which only 5 had facilities for performing TARE) and the SARAH trial (which was carried out at 25 centres); therefore, some contributing centres with little or only an initial experience, could have artificially worsened the results. Furthermore, considering that TARE is a form of brachytherapy, inexperience also influenced its significant impact on the dosimetric aspects; the tumour-absorbed dose and the liver-absorbed dose were not analysed, even if several studies have demonstrated a clear relationship between tumour dose and response rates [26]. In fact, TARE presents very different technical aspects as compared to TACE; while the latter has been a consolidated technique for many years which does not include dosimetric and physical aspects, TARE, being an intra-arterial brachytherapy, involves not only interventional radiologists but also physicists, doctors of nuclear medicine and radiotherapists, which makes it a much more complex therapy as compared to TACE. All these aspects represented the background of the present study which provided several results deserving appropriate discussion.

In the Authors’ experience, before 2012, median OS was 11.2 months and, from 2012, it was 25.7 months. The Authors prospectively selected more patients with ECOG 0, unilobar disease, a reduced tumour burden and, in particular, those having a minor extension of MaVI. All these features converged for better survival with a key role played by MaVI. Already in 2008, L. Kulik et al. [27] had reported an OS of 4.4 months in patients with main portal vein involvement, of 9.9 months in those with branch involvement and of 15.4 months in patients without MaVI; similar results were obtained by Sangro et al. in 2011 [7]. Moreover, the Milan experience [11, 25] confirmed the extension of MaVI, tumour burden and liver function as reliable predictors of OS [28,29]. It should be noted that the group of patients having all negative prognostic factors had a median OS of 7.8 months. In the SIRveNIB trial and the SARAH trial, the median OS in the TARE groups was 8.8 and 8.0 months, respectively, thus very similar to the worst scenario. Considering all these aspects, the results of the present study showed that, in a population without Vp4 patients, an improvement in OS over time was achieved for the Vp best known to respond to TARE [7,11,25,27], and that no improvement was achieved for more advanced liver dysfunction (namely, ALBI B/C) or for patients with a larger tumour burden. These two latter aspects pointed out that, in these patients, no adjunctive survival benefit could be obtained from TARE refinement (ceiling effect).

Experience does not only regard patient selection, but also both embolization technique and dosimetry. These two aspects would have been difficult to estimate without the statistical approach used herein. Once the clinical and radiological features became identical in the two periods, all subsequent differences should have been related to improvement in the treatment itself. Entropy balance allowed estimating the HRs derived from this improvement and, after balancing all known confounders, being treated in the most recent years provided an HR of 0.53 (95% CI: 0.41–0.67). The Authors felt that this was the consequence of two factors. First, the progressive transition from lobar treatment to segmental embolization which is known to be more effective than lobar treatment [30]. Second, and probably the most important, the introduction, after 2011, of personalised dosimetry [31]. The transition from delivered activity approximated using the BSA formula (which does not account for adsorbed doses to the tumour, liver, and lungs) to personalised dosage using the partition model optimised the delivered activity itself. It allowed boosting and optimising the dose delivered to the tumour while reducing potential hepatic and pulmonary toxicity, as shown by the reduction in adverse events in the most recent years. The present results showed that, once covariate distributions were adequately handled, the median survival of patients without personalised dosimetry was very similar to those who received <120 Gy, and that patients who received ≥120 Gy had improved survival with respect to both groups. These findings confirmed the subanalysis of the SARAH trial [32] and strongly supported the expert recommendation to perform TARE with the aim of obtaining an adsorbed dose of at least 100–120 Gy, concluding that personalised activity prescription, based on dosimetry and multidisciplinary management, should be mandatory for its optimisation [33].

The present study also analysed the Impact of subsequent therapies. This was necessary to establish whether and what type of combination of therapies could improve survival. The time-dependent approach adopted herein overcame “survivor treatment selection bias”, a specific type of time-dependent bias occurring in survival analyses whereby patients who live longer are often more likely to receive treatment than patients who die early. First, after reweighting, surgical therapies, TACE, and sorafenib were equally distributed among the two periods; thus, the survival benefit of the most recent period was consequent to the improved efficacy and not to the adoption of these different treatments. Second, and more importantly, only surgery provided an additional survival advantage. This aspect demonstrated that, from an intention-to-treat point of view, TARE is suitable as downstaging or conversion therapy prior to subsequent potentially curative approaches [34]. Conversely, sorafenib did not have any additional benefit. In the past, some authors had already investigated the potential effect of sorafenib in addition to TARE [35,36], and the present results are in line with the most robust findings resulting from the SORAMIC trial which showed no improved survival when combining these therapies. However, considering the recent results of immunotherapy in the treatment of HCC, which show a superiority of the latter as compared to sorafenib, it would be desirable to evaluate the efficacy of TARE combined with immunotherapy. Multiple studies are underway and the results of these studies are awaited [37].

The present study suffered from the limitations intrinsic to retrospective studies. However, the establishment of a learning curve and the evaluation of technical improvements can only be carried out post hoc. Despite this, the present data were derived from a prospectively collected database; thus, these were the best data obtainable to evaluate the aims of this study. Another limitation could be that this was a single-centre study; however, the Authors believe that it could pave the way for future studies aimed at establishing benchmark values from expert centres worldwide. The latter aspect is necessary for future trial designs with the aim of optimising TARE treatment when evaluating it against a comparator, such as the SARAH and the SIRveNIb trials.

In conclusion, the present study showed that better patient selection and, more importantly, refinement of the technique and the adoption of personalised dosimetry greatly improved survival expectations after TARE for unresectable HCC. This could allow more patients to undergo subsequent potentially curative surgery, the only treatment which could additionally prolong survival. On the contrary, sorafenib administered after TARE did not impact life-expectancy.

## Figures and Tables

**Figure 1 jcm-11-07469-f001:**
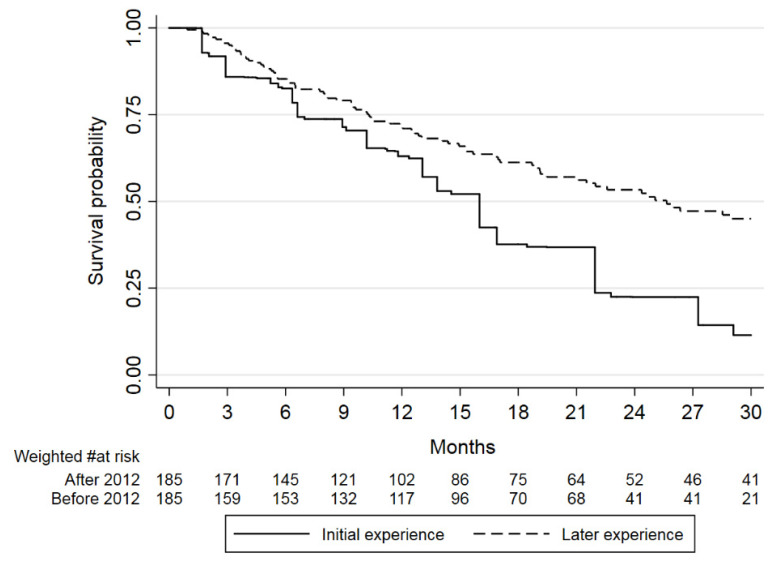
Overall survival in the reweighted population treated before 2012 and in the patients treated in the period from 2012–2020. These survival curves resulted from the balance of the baseline features and subsequent therapies reported in Table 2. The number of patients before 2012 is derived from the mathematical calculation of the weights generated using entropy balance. The HR was 0.53 (95% CI: 0.41–0.67) in favor of the most recent period (*p* = 0.001).

**Figure 2 jcm-11-07469-f002:**
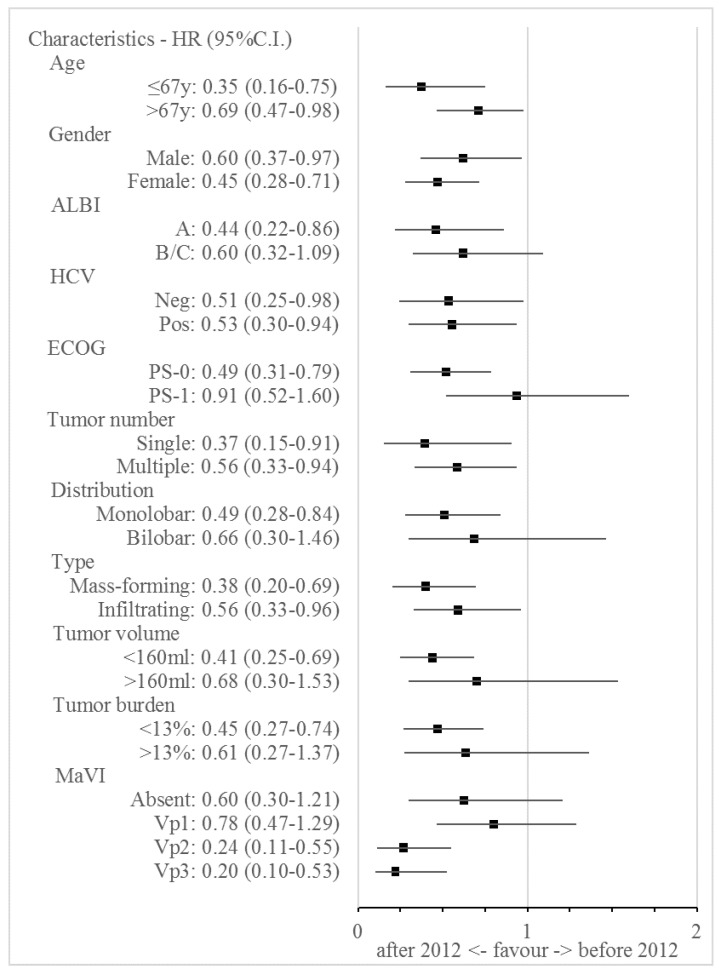
The results of the interaction between the clinical and the radiological characteristics, and the time of treatment in the reweighted study population. The weighted population size of the patients treated before 2012 is equal to that of the patients treated from 2012 onward (*n* = 185).

**Table 1 jcm-11-07469-t001:** The clinical and radiological characteristics of patients treated with radioembolisation divided by period. The continuous variables were reported as median and interquartile range (IQR) (25th–75th percentiles) and were compared using the Kruskal–Wallis test. The categorical variables were compared using the chi-square test or the Fisher’s exact test. The majority of patients underwent only one TARE procedure (*n* = 225; 88.9%), the percentage being similar before 2012 (*n* = 60; 88.2%) and after 2012 (*n* = 165; 89.2%; *p* = 0.823).

	2005–2011 (*n* = 68)	2012–2020 (*n* = 185)	*p*
Age (years)	67 (60–73)	67 (56–74)	0.937
Male	55 (80.9%)	149 (80.5%)	1.000
Hepatitis C	41 (60.3%)	102 (55.1%)	0.478
ECOG–PS of 1	11 (16.2%)	12 (6.5%)	0.025
Unsuccessful TACE	32 (47.1%)	60 (32.4%)	0.039
Single tumour	21 (30.9%)	75 (40.5%)	0.189
ALBI class			0.311
A	26 (38.2%)	68 (36.8%)	
B	41 (60.3%)	117 (63.2%)	
C	1 (1.5%)	0 (0.0%)	
AFP (log10; ng/mL)	1.94 (1.12–3.31)	1.72 (0.84–3.01)	0.147
Bilobar involvement	32 (47.1%)	47 (25.4%)	0.001
Infiltrative type	22 (32.4%)	74 (40.0%)	0.307
Target Tumour volume (cm^3^)	193.2 (96.1–295.9)	153.5 (82.0–331.8)	0.631
Tumour burden (% of liver volume)	19.2 (10.7–35.0)	11.1 (6.3–22.7)	0.006
Presence of MaVI	35 (51.5%)	102 (55.4%)	0.670
Degree of MaVI			0.019
Vp1	19/35 (54.3%)	39/102 (38.2%)	
Vp2	7/35 (20.0%)	47/102 (46.1%)	
Vp3	9/35 (25.7%)	16/102 (15.7%)	
Injected dose (GBq)	1.66 (1.31–1.89)	1.40 (1.00–1.83)	0.001
Tumour adsorbed dose (Gy)	NA	270.3 (152.7–438.5)	-
≥100 Gy	NA	174 (94.1%)	-
≥120 Gy	NA	165 (89.2%)	-
Subsequent therapies	24 (35.3%)	97 (52.4%)	0.016
Liver transplantation	2 (2.9%)	8 (4.3%)	1.000
Hepatic resection	2 (2.9%)	6 (3.2%)	1.000
TACE	15 (22.1%)	41 (22.2%)	1.000
Sorafenib	11 (16.2%)	62 (33.5%)	0.008

Abbreviations: TACE: transarterial chemoembolisation; ECOG-PS:Eastern Cooperative Oncology Group-Performance Status; ALBI: Albumin–Bilirubin; AFP: alpha-fetoprotein; MaVI: macroscopic (neoplastic) vascular invasion; GBq: Giga Becquerel; Gy: Gray; Vp1: portal vein 1.

**Table 2 jcm-11-07469-t002:** The distribution of clinical and radiological characteristics after entropy balance reweighting.

	2005–2011 (*n* = 68 *)	2012–2020 (*n* = 185)	D †
Age > 67 years	35 (51.4%)	95 (51.3%)	0.001
Male	60 (89.0%)	153 (82.9%)	0.007
Hepatitis C	37 (55.1%)	102 (55.2%)	0.002
ECOG–PS of 1	4 (6.5%)	12 (6.5%)	0.001
Unsuccessful TACE	23 (33.3%)	59 (31.8%)	0.003
Single tumour	28 (40.5%)	75 (40.5%)	0.001
ALBI class B/C	43 (63.2%)	117 (63.2%)	0.001
AFP (log10; ng/mL)	29 (43.2%)	80 (43.2%)	0.001
Bilobar involvement	17 (25.4%)	47 (25.5%)	0.001
Infiltrative type	27 (40.0%)	74 (40.0%)	0.001
Tumour volume > 160 cm^3^	32 (47.6%)	86 (46.5%)	0.002
Tumour burden > 13%	30 (45.4%)	86 (46.5%)	0.002
Macrovascular invasion			
Absent	31 (44.9%)	83 (44.9%)	0.001
Vp1	14 (21.1%)	39 (21.1%)	0.001
Vp2	17 (25.4%)	47 (25.4%)	0.001
Vp3	6 (8.6%)	16 (8.6%)	0.001
Injected dose > 1.5 GBq	26 (38.4%)	71 (38.4%)	0.001
Main subsequent therapy adopted			
Resection or transplantation	5 (7.6%)	14 (7.6%)	0.001
TACE	14 (20.0%)	37 (20.0%)	0.001
Sorafenib	17 (24.9%)	46 (24.8%)	0.001

* Number of patients derived from the mathematical calculation of the weights generated using entropy balance. † Standardised differences (d-values) indicate the magnitude of the difference. Values < 0.1 commonly indicate negligible differences between the two groups. In the present analysis, the values were always < 0.01, indicating that the balance was almost perfect. Abbreviations: TACE: transarterial chemoembolisation; ECOG-PS: Eastern Cooperative Oncology Group-Performance Status; ALBI: Albumin—Bilirubin; AFP: alpha-fetoprotein; GBq: Giga Becquerel; Vp1: portal vein 1.

**Table 3 jcm-11-07469-t003:** Occurrence of adverse events within 1 month from transarterial radioembolisation, divided by treatment period, and before and after entropy balance reweighting. This table points out what the mortality and morbidity of patients treated in the first period would have been with the increased expertise and personalised dosimetry of the more recent period.

	Before Weighting		After Weighting		Comparator
	2005–2011 (*n* = 68)	*p* †	2005–2011 (*n* = 68)	*p* †	2012–2020 (*n* = 185)
Mortality	0 (0.0%)	1.000	0 (0.0%)	1.000	1 (0.5%)
Any AE	23 (33.8%)	0.112	15 (22.1%)	0.867	44 (23.8%)
Ascites	13 (19.1%)	0.321	8 (11.7%)	0.834	25 (13.5%)
I-II	9 (13.2%)	0.666	6 (8.8%)	0.652	21 (11.3%)
III-IV	4 (5.9%)	0.216	2 (2.9%)	0.661	4 (2.2%)
Pulmonary events	6 (8.8%)	0.026	2 (2.6%)	0.661	4 (2.2%)
I-II	2 (2.9%)	0.613	0 (0.0%)	0.566	3 (1.6%)
III-IV	4 (5.9%)	0.019	1 (1.5%)	0.466	1 (0.5%)
RILD III-IV	5 (7.4%)	0.016	2 (2.6%)	0.293	2 (1.1%)
Fever	4 (5.9%)	0.086	1 (1.5%)	1.000	3 (1.6%)
I-II	3 (4.4%)	0.348	1 (1.5%)	1.000	3 (1.6%)
III-IV	1 (1.5%)	0.269	0 (0.0%)	1.000	0 (0.0%)
Cholecystitis III-IV	3 (4.4%)	0.019	0 (0.0%)	1.000	0 (0.0%)
Fatigue	2 (2.9%)	0.112	5 (7.4%)	0.803	17 (9.2%)
I-II	0 (0.0%)	0.039	4 (5.9%)	1.000	11 (5.9%)
III	2 (2.9%)	0.177	0 (0.0%)	1.000	1 (0.5%)
GI events III-IV	0 (0.0%)	1.000	1 (1.5%)	0.466	1 (0.5%)

† Values referred to the comparator (2012–2020). Abbreviations: AE: adverse event; RILD: radiation-induced liver disease; GI: gastrointestinal.

## Data Availability

The data presented in this study are available on request from the corresponding author. The data are not publicly available due to ethical and privacy restrictions.

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
