# Peer review of "Improved Survival after Transarterial Radioembolisation for Hepatocellular Carcinoma Gives the Procedure Added Value"

_jcm, 2022, doi:10.3390/jcm11247469_

Round 1
Reviewer 1 Report
Thanks for the opportunity to review this retrospective study investigating possibilities to improve TARE in HCC.
The study is well written and data presentation is very good.
As most studies in advanced HCC, data derrives from the time of Sorafenib as the only approved systemic treatment. Yet, we are heading towards a new era of immunotherapy. Although the study data is older, recent results of studies such as IMBrave150 must be mentioned as it must at least be assumed that subsequent immunotherapy after TARE will further prolong survival compared to Sorafenib.
This is fortunately the only aspect missing in this study. Congratulations!
Author Response
"Please see the attachment."

Reviewer 2 Report
Final comments:
This paper shows that transarterial radioembolisation (TARE) is one of the good strategies for advanced HCC patients.Their data looks reasonable and conclusions encourage for patients who could not stop HCC growth by other treatment.
One thing, I would like to know is that are there any technical differences between TACE and TARE. Please discuss this point at discussion.
